# Gut Microbiome and Metabolome Alterations in Overweight or Obese Adult Population after Weight-Loss *Bifidobacterium breve* BBr60 Intervention: A Randomized Controlled Trial

**DOI:** 10.3390/ijms252010871

**Published:** 2024-10-10

**Authors:** Zhouya Bai, Ying Wu, Dejiao Gao, Yao Dong, Yujia Pan, Shaobin Gu

**Affiliations:** 1College of Food and Bioengineering, Henan University of Science and Technology, Luoyang 471000, China; spbaizhouya@163.com (Z.B.); wuying2000@126.com (Y.W.); gaodej522@163.com (D.G.); 17739736258@163.com (Y.P.); 2Henan Engineering Research Center of Food Microbiology, Luoyang 471000, China; 3Germline Stem Cells and Microenvironment Laboratory, College of Animal Science and Technology, Nanjing Agricultural University, Nanjing 210095, China; yao.dong@stu.njau.edu.cn

**Keywords:** *Bifidobacterium breve* BBr60, overweight or obese, randomized controlled trial, gut microbiota, metabolic profile

## Abstract

Probiotics, known for regulating gut microbiota, may aid those with overweight or obesity, but their mechanisms require more research. This study involved 75 overweight or obese young adults, randomly assigned to either a *Bifidobacterium breve* BBr60 (BBr60) group or a placebo group. Both groups received diet guidance and took either BBr60 (1 × 10^10^ CFU/day) or a placebo for 12 weeks. Researchers analyzed body composition, serum glucose, lipids, liver and kidney function, comprehensive metabolome, and intestinal homeostasis before and after the intervention. After 12 weeks, BBr60 significantly reduced weight and BMI compared to pretreatment levels and outperformed the placebo. The BBr60 group also showed improved blood biochemistry, with notably lower fasting blood glucose (FBG) levels than the placebo group (*p* < 0.05). Additionally, BBr60 influenced vital serum and fecal metabolites related to three amino acid metabolic pathways and regulated the bacteria *Dialister*, *Klebsiella*, and *Bacteroides*, which correlated strongly with serum metabolites. These findings indicate that BBr60 can safely and effectively regulate BMI, body weight, serum glucose, lipids, and liver function markers, which may involve BBr60’s impact on key gut bacteria, which influence metabolites related to the valine, leucine, and isoleucine biosynthesis; glycine, serine, and threonine metabolism; and alanine, aspartate, and glutamate metabolism.

## 1. Introduction

The global obesity epidemic has emerged as a significant public health concern due to the prevalence of poor dietary choices and sedentary lifestyles in conjunction with social advancements and improved living conditions [1]. According to the World Health Organization (WHO), approximately 2.5 billion adults worldwide were classified as overweight (body mass index, BMI ≥ 25) in 2022, with 890 million individuals falling into the category of obesity (BMI ≥ 30). This equates to 43% of adults being overweight and 16% living with obesity [2]. In the past twenty years, there has been a significant increase in the prevalence of both general and abdominal obesity among adults of diverse age groups and genders in China [3]. The detrimental impacts of overweight or obesity extend to various bodily functions, leading to the development of numerous comorbidities such as hyperlipidemia, type 2 diabetes, hypertension, and other disorders [4]. Increasing evidence suggests that the initiation and progression of overweight/obesity are closely associated with the homeostatic regulation of overall metabolism. Especially, microbial metabolites, as the primary way that gut microbes interact with hosts, are important for maintaining the metabolic homeostasis of the organism [5]. According to the existing evidence, identified microbial metabolic signatures have been associated with obesity [6], such as short-chain fatty acids (SCFAs) [7], trimethylamine N-oxide (TMAO) [8], bile acids [9], linoleic acid [10], and branched-chain fatty acids (BCFAs). The diverse metabolic products generated by the gut microbiota permeate into the intestinal mucosa, subtly influencing intestinal cellular activities. A subset of these metabolites possesses the capability to traverse the intestinal barrier, subsequently disseminating to distant organs through either circulatory routes or neural pathways [11].

Furthermore, overweight/obesity-associated microbial metabolites are regulated by the intestinal microbiota to affect the metabolic health of overweight/obese [12]. Multiple studies have demonstrated a notable disparity in the composition of intestinal bacteria between individuals of normal weight and those with overweight/obesity [13]. The alterations in microbiome associated with overweight/obesity, commonly referred to as ‘dysbiosis’, are characterized by a deficiency in beneficial functions or the prevalence of detrimental microbial activity [14]. A lower proportion of *Bifidobacteria*, *Faecalibacterium prausnitzii*, *Akkermansia muciniphila*, *F. prausnitzii*, which are considered as the benefic strains, were found in overweight and obesity [15,16].

Probiotics are acknowledged as a valuable supplementary approach for regulating microbiota dysbiosis, consisting of viable micro-organisms that can successfully modify imbalanced microbiota to enhance key obesity-related factors when given in precise doses of probiotic strains [17]. Additionally, previous studies have identified *Bifidobacterium*, *Lactobacillus*, *Enterococcus*, *Streptococcus*, and *Saccharomyces*, as well as *E. coli* Nissle 1917 and the yeast *Saccharomyces boulardii*, as the primary probiotic genera [18,19]. Especially, *Bifidobacterium* species are frequently utilized as functional foods and dietary supplements, purportedly aiding in the prevention of dysmetabolic diseases through enhancement of metabolic function [20]. Furthermore, *Bifidobacterium* is one of the main producers of SCFAs in the intestinal flora to participate in regulating energy homeostasis [21]. As one kind of *Bifidobacterium*, the commercially available probiotic *Bifidobacterium breve* BBr60 (BBr60) has demonstrated anti-inflammatory and antioxidant properties. Given its potential for addressing metabolic disorders, further research is needed to elucidate the mechanisms by which *Bifidobacterium breve* BBr60 intervenes in metabolic disturbances associated with overweight/obesity, as well as to explore its potential application in clinical settings.

Therefore, we conducted a double-blind, randomized, placebo-controlled trial to examine the comparative efficacy and underlying mechanisms of *Bifidobacterium breve* BBr60 in modulating disrupted metabolic profiles and gut microbiota among overweight and obese young individuals in China, which can provide a scientific basis for the dietary and clinical application of *Bifidobacterium breve* BBr60 in the prevention and management of overweight and obesity.

## 2. Results

### 2.1. Baseline Characteristics

Table 1 presents the baseline characteristics of the participants, including sex distribution, mean age, baseline BMI values, and waist-to-hip ratio (WHR). The distribution of sex did not show a significant difference between the BBr60 group, where 72.7% (n = 24) were female, and the placebo group, where 59.4% (n = 19) were female. The mean age was 27.88 ± 8.65 years in the BBr60 group and 30.38 ± 8.45 years in the placebo group. The baseline BMI values were 30.80 ± 3.21 kg/m^2^ and 31.96 ± 2.95 kg/m^2^, respectively. The WHR also did not exhibit a significant difference between groups, with values of 0.99 ± 0.05% in the BBr60 group and 1.01 ± 0.05% in the placebo group. There were no significant differences observed in lipid levels, renal and hepatic function indicators, anthropometric and body composition data, and biochemical parameters between the two groups.

### 2.2. Primary Outcome

#### Efficacy of BBr60 on Weight and BMI

The study measured the change in weight as a secondary outcome by comparing baseline weight to weight at 12 weeks. The BBr60 group experienced a statistically significant decrease in weight of 4.67 (*p* < 0.0001) compared to a decrease of 2.82 in the placebo group (*p* = 0.0006), and the difference in body weight between 0 weeks and 12 weeks in the BBr60 group was significantly higher than that of the placebo group (*p* = 0.047). Additionally, after 12 weeks, the BMI value in the BBr60 group was 86.19, which was 4.55 lower than the placebo group (*p* = 0.1137), as shown in Table 2 and Figure 1A.

The primary outcome of this study was the change in BMI, which was calculated as the difference in BMI from baseline to 12 weeks. The BMI significantly decreased with the BBr60 intervention (*p* < 0.0001) and also decreased in the placebo group (*p* = 0.0019). At the end of the 12-week period, the BMI value was 29.32 in the BBr60 group, which was lower than the placebo group (*p* = 0.056) (Table 2, Figure 1B).

### 2.3. Secondary Outcomes

#### 2.3.1. Efficacy of BBr60 on BFP and WHR

BFP and WHR are commonly used in the clinical evaluation of body fat distribution [22]. In the current study, BFP and WHR were significantly decreased in BBr60 and placebo groups after a 12-week intervention. Notably, BBr60 intervention presented a more obvious effect than placebo, with a lower *p*-value (*p* < 0.0001). Additionally, the values of BFP and WHR in the BBr60 group were both lower than those of the placebo group in 12 weeks (Table 2, Figure 1C,D).

#### 2.3.2. Efficacy of BBr60 on Blood Glucose and Lipid

In the BBr60 group, there was a significant reduction in fasting blood glucose (FBG) from a baseline value of 5.87 (SD 0.45) to 5.26 (SD 0.57) after the 12-week intervention period (Table 3). Additionally, the FBG levels in the placebo group also showed a significant decrease; however, the mean FBG level in the BBr60 group was significantly lower than that of the placebo group after the 12-week intervention (*p* = 0.0381).

No clinically meaningful differences between BBr60 and placebo groups after a 12-week intervention were noted in plasma lipids, including TC, TG, HDL-C, and HDL-C. Notably, HDL-C and LDL-C were significantly regulated by BBr60 intervention with *p* < 0.0001. Furthermore, the level of HDL-C of the BBr60 group was higher than that of the placebo group, and LDL-C of the BBr60 group was lower than that of the placebo group at week 12.

#### 2.3.3. Efficacy of BBr60 on Liver Function and Renal Function

Liver function (TP, ALB, GLB, ALT, and AST) and kidney function (BUN, CRE-E) were measured at week 12 (Table 4). Additionally, the levels of ALB, GLB, ALT, and AST were significantly reduced following BBr60 intervention, with levels in the BBr60 group being lower than those in the placebo group at week 12. No significant differences were observed for TP and BUN.

#### 2.3.4. Efficacy of BBr60 on Metabolic Pathway

##### Alteration of Serum Metabolism after 12 Weeks of BBr60 Intervention

The aforementioned study demonstrated that, under the condition of effective regulation in the placebo group, which may be largely attributed to daily healthy diet recommendations, BBr60 significantly improved clinic indexes of obesity, especially FBG and weight. Additionally, we explored the regulatory effects of BBr60 on serum metabolism in individuals who are overweight or obese. OPLS-DA was employed to identify metabolites that met the criterion of variable influence on projection (VIP) > 1 between the BBr60-before and BBr60-after groups. The samples from BBr60-before and BBr60-after groups were distinctly separated in the OPLS-DA score plot (Figure 2A), with model parameters R^2^Y = 0.50 and Q^2^ = −0.89 as shown in Figure 2B. PCA was employed to evaluate the alterations in metabolic profiling between the BBr60-before and BBr60-after groups (Figure 2C). Although no distinct separation was observed, a substantial number of significantly regulated fecal metabolites (*p* < 0.05, VIP > 1) were identified in the volcano plot (Figure 2D). Specifically, 292 metabolites were significantly upregulated, while 351 serum metabolites were downregulated in the BBr60-before group compared to the BBr60-after group. Detailed information regarding these metabolites is provided in Appendix A.

Furthermore, the analysis of 42 significant metabolic pathways (*p* < 0.05) associated with significant fecal metabolites (VIP > 1, *p* < 0.05 between BBr60-before and BBr60-after groups) was conducted (see Appendix A, and Figure 2E,F). The top 15 KEGG pathways (*p* < 0.001) exhibited alterations in response to BBr60 treatment for 12-week intervention (Figure 2E). These pathways include central carbon metabolism in cancer, protein digestion and absorption, biosynthesis of amino acids, glyoxylate and dicarboxylate metabolism, pantothenate and CoA biosynthesis, citrate cycle (TCA cycle), carbon metabolism, valine, leucine and isoleucine biosynthesis, glycine, serine and threonine metabolism, aminoacyl-tRNA biosynthesis, 2-oxocarboxylic acid metabolism, alanine, aspartate and glutamate metabolism, mineral absorption, ABC transporters, and beta-alanine metabolism (Figure 2F). Biosynthesis of amino acids, carbon metabolism, and 2-oxocarboxylic acid metabolism categorized under global and overview maps, exhibited significant up-regulation in 15.69%, 11.76, and 12.75% of differential metabolites, respectively, when influenced by BBr60. Furthermore, the significantly regulated pathways associated with valine, leucine and isoleucine biosynthesis, glycine, serine and threonine metabolism as well as alanine, aspartate and glutamate metabolism, were all categorized under amino acid metabolism with 5.88%, 7.84%, and 5.88% of differential metabolites, respectively. The pathways of glyoxylate and dicarboxylate metabolism, as well as citrate cycle (TCA cycle), exhibited 9.8% and 5.88% differential metabolites among all metabolites in the pathway, which were significantly regulated by BBr60. These pathways are categorized under carbohydrate metabolism. In serum metabolism, 643 metabolites regulated by BBr60 were associated with 42 significant metabolic pathways. Notably, the pathways of glyoxylate and dicarboxylate metabolism, TCA cycle, valine, leucine and isoleucine biosynthesis, glycine, serine and threonine metabolism, as well as alanine, aspartate, and glutamate metabolism warrant further investigation.

##### Alteration of Fecal Metabolism after 12 Weeks of BBr60 Intervention

To analyze the underlying mechanisms of BBr60’s effective intervention on overweight or obesity, fecal metabolism was examined before and after BBr60 treatment. Orthogonal partial least squares discriminant analysis (OPLS-DA) was utilized to identify metabolites with a variable influence on projection (VIP) score greater than 1, both prior to and following the BBr60 intervention. The OPLS-DA score plot (Figure 3A) demonstrated a clear separation between the BBr60-before and BBr60-after groups, with model parameters R^2^Y = 0.75 and Q^2^ = −0.71, as depicted in Figure 3B. PCA was utilized to assess the changes in metabolic profiles between the BBr60-before and BBr60-after groups (Figure 3C). Despite the absence of a clear separation, 514 fecal metabolites exhibited significant regulation (*p* < 0.05, VIP > 1), as illustrated in the volcano plot (Figure 3D) and detailed in Appendix A. Of these, 225 metabolites were significantly upregulated, whereas 289 metabolites were downregulated following the 12-week BBr60 intervention.

Additionally, a comprehensive analysis of the metabolic pathways associated with these significantly altered fecal metabolites (VIP > 1, *p* < 0.05 between BBr60-before and BBr60-after groups) was performed (Appendix A, and Figure 4). A total of 52 metabolic pathways exhibited alterations in response to BBr60 treatment compared to the pretreatment condition (Appendix A). Among these pathways, 25 demonstrated statistical significance (*p* < 0.05). Notably, 15 pathways, which were downregulated by BBr60 over a 12-week period, met a more stringent significance threshold (*p* < 0.01) and are illustrated in Figure 4A. These pathways encompass ABC transporters, mineral absorption, central carbon metabolism in cancer, biosynthesis of amino acids, valine, leucine and isoleucine biosynthesis, nucleotide metabolism, protein digestion and absorption, glycine, serine and threonine metabolism, aminoacyl-tRNA biosynthesis, 2-oxocarboxylic acid metabolism, linoleic acid metabolism, pyrimidine metabolism, cysteine and methionine metabolism, valine, leucine and isoleucine degradation, and glycerophospholipid metabolism (Figure 4B). Biosynthesis of amino acids, nucleotide metabolism, and 2-oxocarboxylic acid metabolism, which are categorized under global and overview maps, exhibited significant down-regulation in 15.38%, 10.77%, and 13.85% of differential metabolites, respectively, when influenced by BBr60. Additionally, four metabolic pathways were classified under amino acid metabolism, including valine, leucine, and isoleucine biosynthesis; glycine, serine, and threonine metabolism; cysteine and methionine metabolism; and valine, leucine, and isoleucine degradation. Furthermore, two metabolic pathways, including linoleic acid metabolism and glycerophospholipid metabolism, were categorized as lipid metabolism.

In fecal metabolism, 514 fecal metabolites exhibited significant regulation (*p* < 0.05, VIP > 1) and related to 25 significant metabolic pathways, including amino acid metabolism and lipid metabolism, in the BBr60-before vs. BBr60-after groups.

#### 2.3.5. Efficacy of BBr60 on Gut Microbiota

The richness, diversity, and evenness of the gut microbiota in the BBr60 placebo groups during the 12th week were assessed using α and β diversity analyses (Figure 5A–J). The α diversity was significantly increased following a 12-week BBr60 intervention, as indicated by metrics such as goods_coverage, pielou_e, and the Shannon index. Additionally, a clear separation of samples from the BBr60-before and BBr60-after groups was observed based on jaccard_distance and unweighted_unifrac_distance. At the phylum level, Firmicutes and Actinobacteria were the predominant species in both the placebo-after and BBr60-after groups, followed by Proteobacteria and Bacteroidota (Figure 6A and Appendix A). As shown in Figure 6B and Appendix A, the dominant bacterial groups in each group are mainly *Bifidobacterium*, *Streptococcus*, *Agathobacter*, *Escherichia-Shigella*, *Faecalibacterium*, *Megamonas*, *Erysipelotrichaceae_UCG-003*, *Dialister* at the genus level. The relative abundance of *Escherichia-Shigella*, *Dialister*, *Phascolarctobacterium*, *Klebsiella*, *Bacteroides*, and *Veillonella* in the BBr60-after group was higher than that of BBr60-before the group (Figure 6C). The 12-week intervention with BBr60 resulted in a statistically significant decrease in the relative abundance of the class Bacilli (*p* < 0.05). These results indicate that BBr60 can effectively increase the abundance of *Escherichia-Shigella*, *Dialister*, *Phascolarctobacterium*, *Klebsiella*, *Bacteroides*, and *Veillonella*, while concomitantly reducing the abundance of the class Bacilli. These microbial species may serve as key indicators in the regulation of obesity and could potentially aid in differentiating between the gut microbiota of obese and normal-weight individuals. Besides, there was no serious adverse event in any of the groups during the study period, suggesting a favorable safety profile for BBr60.

### 2.4. Correlation Analysis

The potential relationships of gut flora, metabolic profiling, and clinic index were conducted by Spearman correlation analysis (Figure 7A–C). *Escherichia-Shigella*, *Dialister*, *Phascolarctobacterium*, *Klebsiella*, *Bacteroides*, and *Veillonella*, the vital genera significantly regulated by BBr60, showed the significantly certain metabolites (Figure 7A,B). Specially, *Dialister* and *Bacteroides* exhibited obvious correlation with most serum metabolites. They both showed the negative relationship with the serum metabolites, including butanoic acid, isovaleric acid, oxalic acid, 3-hydroxypyruvic acid, malonic acid, o-acetylserine, valine, beta-alanine, ketoleucine, as well as, exhibited positive relationship with glutamate, serine, asparagine (Figure 7A). Additionally, those metabolites were involved in 14 pathways of 2-oxocarboxylic acid metabolism, ABC transporters, alanine, aspartate and glutamate metabolism, aminoacyl-tRNA biosynthesis, beta-alanine metabolism, biosynthesis of amino acids, carbon metabolism, central carbon metabolism in cancer, glyoxylate and dicarboxylate metabolism, mineral absorption, pantothenate and CoA biosynthesis, protein digestion and absorption, valine, leucine and isoleucine biosynthesis, glycine, serine, and threonine metabolism. A total of 3 out of 14 were categorized under global and overview maps, including biosynthesis of amino acids, 2-oxocarboxylic acid metabolism, and carbon metabolism. Furthermore, 3 out of 11 were categorized under amino acid metabolism, including valine, leucine and isoleucine biosynthesis, glycine, serine, and threonine metabolism, as well as alanine, aspartate and glutamate metabolism. Additionally, three pathways of amino acid metabolism were also significantly regulated by BBr60 on fecal metabolism. Additionally, among metabolites significantly correlated with *Dialister* and *Bacteroides*, five amino acids (asparagine, serine, glutamate, ketoleucine, and valine), which were associated with three pathways of amino acid metabolism, exhibited significant relationships with certain clinic indexes of obesity, such as FBG, ALB, LDL-C, and HDL-C (Figure 7C).

## 3. Discussion

Obesity is a multifaceted condition with a complex pathogenesis involving socioeconomic, hormonal, and neuronal mechanisms, as well as unhealthy lifestyle choices and genetic and epigenetic factors [23]. The World Health Organization predicts that by 2035, 39% of the global adult population will be affected by obesity [2]. Research suggests that obesity and related metabolic disorders are linked to changes in gut microbiota function and composition, which play a significant role in regulating the body’s energy metabolism [24]. Furthermore, alterations in the gut microbiota composition have been linked to the onset of obesity and its related metabolic conditions [25]. Utilizing probiotics to manipulate the gut microbiome may serve as a potential approach for managing metabolic syndrome and obesity-associated complications, such as dyslipidemia and insulin resistance [26]. Nevertheless, the efficacy of probiotics is contingent upon the specific species and dosages employed, as well as the underlying medical condition [27]. Therefore, the current study investigated the effectiveness of BBr60 (1 × 10^10^ CFU, once a day) for 12 weeks on body composition, serum glucose, lipid, liver, and kidney functions in an overweight or obese adult population with BMI ≥ 28 kg/m^2^. After a 12-week intervention, BMI was significantly decreased by BBr60 intervention, with levels in the BBr60 group being lower than that in the placebo group at week 12. Similar trends were seen for body weight, BFP, WHR, FBG, HDL-C, LDL-C, and liver function indexes (ALB, GLB, ALT, AST) in the BBr60 group.

*Bifidobacterium* is a typical probiotic with the ability of reducing intestinal lipopolysaccharide and fortifying intestinal barrier function and has been widely used as probiotic preparations for the treatment of intestinal microecological disorders [28,29]. *Bifidobacterium breve* BBr60 is one kind of *Bifidobacterium*, having been commercialized for anti-inflammatory and antioxidant properties. In this study, BBr60 presented the effectiveness on BMI, body weight, BFP, WHR after a 12-week intervention, and scientific evidence also suggests that alterations in the gut microbiota through the use of probiotics may play a role in changes in body weight and composition [30]. Specifically, administering *Bifidobacterium* to individuals with overweight or obesity (BMI > 24.9 kg/m^2^) resulted in significant reductions in body fat mass (*p* = 0.006), body fat percentage (*p* = 0.02), waist circumference (*p* < 0.00001), and visceral fat area (*p* = 0.003) [31]. A meta-analysis of 15 studies on probiotics demonstrated significant changes in body weight and body fat among obese individuals with a BMI exceeding 25 kg/m^2^, with an average weight loss of 0.6 kg and a BMI reduction of 0.27 kg/m^2^ [32]. The admonition of *Bifidobacterium breve* B-3 (20 billion CFU/day) for 12 weeks significantly reduced the body fat in preobese adults without any adverse effects [33]. In addition, the reducing weight, BMI, BPF, and WHR in the placebo group may be associated with the increased awareness of weight loss by dietary recommendations once a week for all the participants during the trial, and similar results were found in many clinical studies [34,35].

Several studies have demonstrated that probiotic intervention may be beneficial in the management of obesity, as well as various metabolic abnormalities such as dysglycemia, insulin resistance, and dyslipidemia [36]. Additionally, probiotic supplementation has been shown to improve fasting blood glucose levels, insulin sensitivity, and hyperlipidemia [37]. In the present study, levels of fasting blood glucose and LDL-C were significantly reduced, and HDL-C was effectively increased following a 12-week intervention with BBr60. Accordingly, previous studies also presented that *Bifidobacteria* supplementation (50 × 10^9^ CFU/day) for 12 weeks effectively ameliorates hyperglycemia, dyslipidemia by decreasing serum LDL, TG, and glycosylated hemoglobin concentration in type 2 diabetic patients [38]. Administration of *Bifidobacterium animalis* IPLA R1 decreased serum insulin levels with no significant variation in FGB and HOMA index in mice of a short-term diet-induced obesity [39].

The liver plays a crucial role in regulating whole-body cholesterol levels, while the kidney is essential for maintaining overall homeostasis [40,41]. Impaired kidney function and liver dysfunction are commonly observed in individuals with severe obesity [42]. Assessment of liver and kidney function serves as a valid measure for evaluating psychological states in obesity. In a study involving overweight and obese adults undergoing a 12-week BBr60 intervention, liver function indicators (TP, ALP, GLB, ALT, and AST) and renal function markers (BUN) were examined to assess the impact of BBr60 supplementation. Furthermore, supplementation with BBr60 led to notable decreases in ALB, GLB, ALT, and AST levels. Previous research has shown that supplementation with *Bifidobacterium breve* resulted in significant reductions in BUN and creatinine levels compared to a placebo group [38]. Additionally, mice that were gavaged with *Bifidobacterium pseudolongum* exhibited a lower liver-to-body weight ratio and reduced serum levels of ALT, AST, and hepatic triglycerides in cases of nonalcoholic fatty liver disease-associated hepatocellular carcinoma [43]. These findings suggest the efficacy of BBr60 in improving certain clinical indicators in overweight/obese individuals.

The deep mechanism was further analyzed after the effective regulation of BBr60 on clinical indicators of obesity. The metabolic status of the organism was investigated by serum and fecal metabolism as well as gut microbiota, and the potential relationship between metabolic profiles and clinical indexes of obesity. A total of 42 significant metabolic pathways associated with 643 serum metabolites and 25 vital metabolic pathways related to 514 fecal metabolites were obviously regulated by BBr60, and the amino acid metabolism pathways of valine, leucine and isoleucine biosynthesis, glycine, serine and threonine metabolism, as well as alanine, aspartate and glutamate metabolism, were regulated in serum metabolism and fecal metabolism. Significant associations between metabolites involved in three pathways and predominant genera or clinic indexes (such as LDI-C and HDL-C) of BBr60 intervention were found. Extensive research demonstrates that distinct metabolism of amino acids has long been recognized as a feature of obesity [44,45] and probiotics can improve intestinal flora imbalance and regulate intestinal microbial metabolites [46,47] to affect amino acid metabolism and other metabolic pathways [48] to further alleviating overweight or obesity [49].

The potential mechanism of probiotics on overweight or obesity has been discussed in previous studies [47]; more research has focused on gut inflammation and lipid metabolism [39]. The enhancement of glucolipid metabolism by probiotics may be linked to a reduction in bacterial lipopolysaccharides (LPSs), which are known to induce inflammation and obesity [50,51]. While the role of lipopolysaccharides in the regulation of obesity through amino acid metabolism remains unclear, numerous studies have identified significant associations between specific probiotic strains and particular amino acids [52]. Host glutamate levels could be influenced by *B. thetaiotaomicron* colonization, which increases the levels of mRNAs encoding glutamate decarboxylase and glutamate transporter in epithelial cells, and the concentration of plasma glutamate reduces by gavage with *B. thetaiotaomicron* in mice [12,53]. Additionally, the abundance of *B. thetaiotaomicron* showed a negative correlation with the circulation of glutamate in a previous study [12]. Consistently, our study revealed that the plasma glutamate levels, which are involved in the alanine, aspartate, and glutamate metabolism pathways, exhibited significant associations with dominant bacterial strains (*Klebsiella*, *Bacteroides*, and *Dialister*) following the BBr60 intervention, as well as with clinical indices such as fasting blood glucose (FBG) and albumin (ALB). Additionally, other amino acids, including asparagine, serine, ketoleucine, and valine, which are implicated in pathways of valine, leucine and isoleucine biosynthesis, glycine, serine, and threonine metabolism, as well as alanine, aspartate and glutamate metabolism, demonstrated significant correlations with the bacterial strains *Dialister* and *Bacteroides* and with specific clinical indices such as low-density lipoprotein cholesterol (LDL-C).

Generally, the effective regulation of FBG, LDL-C, HDL-C, and ALB may be attributed to the regulation of BBr60 on vital genus (*Klebsiella*, *Bacteroides*, and *Dialişter*) to affect the vital metabolites associated with the pathways of biosynthesis pathways of valine, leucine, and isoleucine, as well as in glycine, serine, and threonine metabolism and alanine, aspartate, and glutamate metabolism. However, although our analysis has suggested potential mechanisms by which BBr60 regulates blood glucose and lipid levels in cases of obesity or overweight, dietary calorie restriction and regular physical exercise may also be significant factors in controlling obesity, as evidenced by the effective biochemical outcomes observed in the placebo group. The amount of physical activity or dietary changes was not collected in detail throughout the study. Therefore, further studies are necessary to investigate the effectiveness of BBr60 in mice, specifically isolating the variables of dietary calorie restriction and daily physical exercise.

## 4. Materials and Methods

### 4.1. Ethics and Informed Consent

This study utilized a randomized, double-blind, placebo-controlled trial design conducted at the School of Food and Bioengineering, Henan University of Science and Technology, from March 2023 to June 2024. The protocol adhered to the World Medical Association Declaration of Helsinki and was approved by the Ethics Committee of the First Affiliated Hospital of Henan University of Science and Technology (NCT06305650).

The PICO statement was the following:

Participants—Patients aged between 19 and 45, having a BMI of 28 kg/m^2^ or higher.

Intervention—Participants were administered daily maltodextrin (3 g) or BBr60 (1 × 10^10^ CFU, once a day) for a duration of 12 weeks. All participants were instructed to reduce their daily energy intake by 1800 kcal and attended a nutrition information course covering topics, such as the risks and causes of overweight and obesity, weight loss principles, dietary recommendations, and rest.

Comparator—Comparison of probiotic and maltodextrin intake in week 12, as well as, before and after probiotic intake.

Outcomes—Primary: changes in body mass index (BMI) from the baseline to 12 weeks after beginning the treatment. Secondary: changes in waist hip ratio (WHR), and body fat rate (BFR) from the baseline to 12 weeks. Other secondary outcomes were the changes in serum biochemical indexes, fecal metabolism, and gut microbiota from baseline to 12 weeks.

### 4.2. Study Design and Population

All participants in the study provided informed consent and met the designated inclusion criteria, which included being between the ages of 19 and 45, having a BMI of 28 kg/m^2^ or higher, and agreeing to participate after being informed about the study procedures and signing a written consent form. The exclusion criteria encompassed various factors that could potentially impact the validity of the results, such as short-term use of objects with similar functions to the test, recent administration of antibiotics, laxatives, or dietary supplements, history of alcohol or drug abuse, presence of serious medical illnesses (e.g., kidney or liver disease, diabetes mellitus, neurological disorders), and pregnancy or lactation without the use of contraception. Studies were required to meet all of these criteria in order to be considered eligible for inclusion in the analysis.

### 4.3. Sample Size and Randomization

A total of 78 participants were screened, with 75 starting treatment and 65 completing the study. Eligible participants were randomly allocated to either the placebo group or the probiotics group using a random number table, as shown in Figure 8. The trial adhered strictly to the initial protocol without modifications. Participants in the placebo group were administered daily maltodextrin (3 g), while those in the BBr60 group consumed daily BBr60 (1 × 10^10^ CFU, once a day, provided by Wecare Probiotics Co., Ltd., Suzhou, China) for a duration of 12 weeks. All participants were instructed to reduce their daily energy intake by 1800 kcal and attended a nutrition information course covering topics such as the risks and causes of overweight and obesity, weight loss principles, dietary recommendations, and rest. Patients, study staff, clinical research associates, and statisticians were blinded to the randomization and study products. The residual products and medicines, and empty packing boxes were recovered.

### 4.4. Primary Outcome and Secondary Outcomes

The primary outcome of this study was the change in weight, body mass index (BMI) from the baseline to 12 weeks after beginning the treatment. The key secondary outcomes were changes in waist-hip ratio (WHR), and body fat rate (BFR) from the baseline to 12 weeks. Other secondary outcomes were the changes in serum biochemical indexes, fecal metabolism, and gut microbiota from the baseline to 12 weeks.

### 4.5. Assessment of Body Composition

All subjects were weighed in light clothing without shoes. Body mass index (BMI), waist-hip ratio (WHR), and body fat rate (BFR) were measured with a body composition analyzer (InBody270, InBody, Tokyo, Japan).

### 4.6. Blood Sample Collection and Biochemical Measurements

Collection and biochemical measurements of blood were carried out at the hospital of Henan University of Science and Technology by clinical standard assays right after fasting blood sampling at baseline, and after 12 weeks of intervention. Blood samples were collected following an overnight fast of at least 10 h for clinical chemistry analyses. Serum samples were subsequently centrifuged and stored at −80 °C until analysis. The following parameters were measured using an automatic biochemical analyzer (KHB ZY-1280, Shanghai Kehua Bio-engineering Corporation, Shanghai, China): fasting blood glucose (FBG), alanine aminotransferase (ALT), aspartate aminotransferase (AST), alkaline phosphatase (ALP), total protein (TP), albumin (ALB), globular proteins (ALP), total bilirubin (TB), blood urea nitrogen (BUN), and lipid profile including total triglycerides (TG), total cholesterol (TC), high-density lipoprotein cholesterol (HDL-C), and low-density lipoprotein cholesterol (LDL-C).

### 4.7. Serum and Fecal Metabolomic Analysis

Fasted serum samples were collected pre- and post-treatment, then centrifuged at 3000× *g* for 15 min. For metabolomics analysis, 100 μL of serum was mixed with 400 μL of extraction solution (MeOH: ACN, 1:1 (*v*/*v*)) containing deuterated internal standards. The mixture was vortexed for 30 s, sonicated for 10 min at 4 °C, and incubated for 1 h at −40 °C to precipitate proteins. Finally, the samples were centrifuged at 12,000 rpm (13,800× *g*) for 15 min at 4 °C. For analysis, the supernatant was transferred into a fresh vial of glass.

Fecal samples were mixed with beads and extraction solution, containing deuterated internal standards, and vortexed for 30 s.

Quality control (QC) samples were prepared by mixing an equal amount of supernatant from each sample. LC-MS/MS analysis was performed using a UHPLC system (Vanquish, Thermo Fisher Scientific, San Jose, CA, USA) coupled to an Orbitrap MS. The mobile phase consisted of 25 mmol/L ammonium acetate and ammonia hydroxide in water. The autosampler was set at 4 °C with an injection volume of 2 μL. The Orbitrap Exploris 120 mass spectrometer (Orbitrap MS, Thermo Fisher Scientific, San Jose, CA, USA) was used in IDA mode with Xcalibur software (v4.4), continuously evaluating the full scan MS spectrum. The acquisition software monitors the full scan MS spectrum continuously in this mode. ESI source conditions include sheath gas flow rate of 50 Arb, Aux gas flow rate of 15 Arb, capillary temperature of 320 °C, full MS resolution of 60,000, MS/MS resolution of 15,000, collision energy settings of 20/30/40 SNCE, and spray voltage of 3.8 kV (positive) or −3.4 kV (negative). The raw data underwent conversion to the mzXML format through the utilization of ProteoWizard (https://proteowizard.sourceforge.io/projects.html (accessed on 31 July 2024)) and were subsequently analyzed using a custom program developed in R and reliant on XCMS for peak detection, extraction, alignment, and integration. Metabolite identification was facilitated through the use of the R package (v3.8.2) and BiotreeDB (v3.0).

### 4.8. Gut Microbiota Analysis

DNA was extracted from fecal using CTAB following the manufacturer’s instructions. The reagent effectively extracted DNA from trace amounts of samples, particularly bacteria. Blank samples were prepared using nuclear-free water. The eluted DNA was stored at −80 °C until PCR measurement. Primers were tagged with specific barcodes for each sample and sequencing universal primers. PCR was conducted in a 25 μL reaction mixture with 25 ng of template DNA, PCR Premix, primers, and water. The prokaryotic 16S fragments were amplified using specific PCR conditions, including denaturation, annealing, and extension steps. The PCR products were verified using agarose gel electrophoresis. During DNA extraction, ultrapure water was used as a negative control to prevent false-positive PCR results. PCR products were purified using AMPure XT beads (Beckman Coulter Genomics, Danvers, MA, USA) and quantified with Qubit (Invitrogen, Waltham, MA, USA). Amplicon pools were prepared for sequencing, and their size and quantity were assessed using Agilent 2100 Bioanalyzer (Agilent, CA, USA) and Library Quantification Kit for Illumina (Kapa Biosciences, Woburn, MA, USA).

Samples were sequenced on an Illumina NovaSeq platform PE250 (Illumina, San Diego, CA, USA) following the manufacturer’s instructions. Paired-end reads were assigned to samples based on their unique barcode, merged using FLASH (v1.2.8), and quality filtered using fqtrim (v0.94). Chimeric sequences were removed using Vsearch software (v2.3.4). The feature table and sequence were obtained after dereplication using DADA2 (v2019.7). Alpha and beta diversity were calculated by randomly normalizing sequences. The feature abundance was normalized using the relative abundance of each sample according to the SILVA classifier. Alpha diversity was analyzed using 5 indices in QIIME2 (https://qiime2.org), while beta diversity was calculated and visualized using the R package. Blast was used for sequence alignment, and the feature sequences were annotated with the SILVA database for each representative sequence. Other diagrams were implemented using the R package (v3.5.2). Blast was used for sequence alignment, and the SILVA database was used for annotation. Other diagrams were created using R package v3.5.2.

### 4.9. Safety Monitoring

Safety outcomes will be evaluated by monitoring vital signs and body weight at each visit. Participants will undergo blood routine tests, liver and kidney function tests, urine routine tests, and physical examinations at baseline and in the 12th week of treatment. Safety outcomes will be assessed through the evaluation of physical examinations, vital signs, hematological analyses, and reported adverse events or serious adverse events.

### 4.10. Statistical Analysis

Statistical analyses were performed using SPSS version 22.0 (SPSS, Chicago, IL, USA) and Graphpad Prism 8.0 (GraphPad Prism Software, San Diego, CA, USA). The normal distribution of the variables was analyzed by Kolmogorov–Smirnov test. Quantitative data following are reported as mean ± standard deviation (SD), with statistical significance assessed using a two-tailed T test for data conforming to a normal distribution. Additionally, data not conforming to a normal distribution is analyzed using nonparametric tests such as the Mann–Whitney U test for between-group comparisons and the Wilcoxon signed-rank test for within-group comparisons. Categorical variables are presented as numbers (%) and compared using the chi-square test. Principal component analysis (PCA), orthogonal partial least-squared discriminant analysis (OPLS-DA) were conducted by the SIMCA software package (V18.0.1, Sartorius Stedim Data Analytics AB, Umea, Sweden). Pathway analysis was performed by databases including KEGG (http://www.genome.jp/kegg/ (accessed on 31 July 2024)) and MetaboAnalyst (http://www.metaboanalyst.ca/ (accessed on 31 July 2024)). KEGG enrichment analysis of differential metabolites was conducted with Fisher’s test, * *p* < 0.05, ** *p* < 0.01. Correlations between two variables were assessed through Spearman correlation analyses with R (v4.2.1), with statistical significance defined as *p* < 0.05 and an alpha level (α) established at 0.05. The obtained *p*-values validate that the discrepancies observed between groups were not a result of random variation, thus bolstering the reliability of our results.

## 5. Conclusions

Collectively, these findings suggest that BBr60 can safely and effectively reduce body weight, BMI, BFP, WHR, FBG, LDL-C, HDL-C, and improve liver function (ALB, GLB, ALT, and AST). Furthermore, the potential mechanism underlying the effective regulation of clinical indices may be attributed to BBr60’s influence on key genera (*Klebsiella*, *Bacteroides*, and *Dialister*), which in turn affect critical metabolites associated with the pathways of valine, leucine and isoleucine biosynthesis, glycine, serine, and threonine metabolism, as well as alanine, aspartate, and glutamate metabolism. Those results will provide scientific evidence supporting the BBrr60 intervention for attenuating obesity or overweight. Future research should focus on elucidating the deeper mechanisms by which BBr60 affects these key genera and specific metabolites to regulate glucolipid metabolism in murine models.

## Figures and Tables

**Figure 1 ijms-25-10871-f001:**
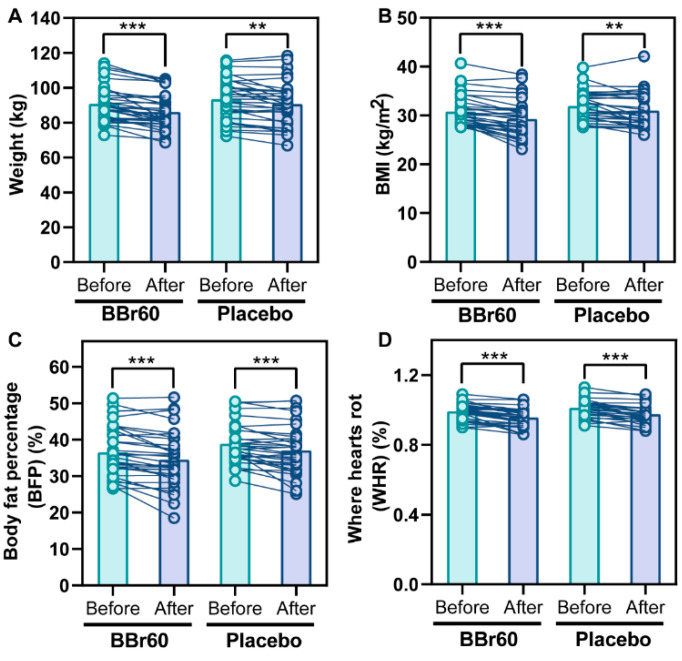
The effect of BBr60 on weight, BMI, and body composition in overweight or obese adult population. (**A**) The effect of BBr60 on weight. (**B**) The effect of BBr60 on BMI. (**C**) The effect of BBr60 on body fat percentage (BFP). (**D**) The effect of BBr60 on waist-to-hip ratio (WHR). ** *p* < 0.001, *** *p* < 0.001.

**Figure 2 ijms-25-10871-f002:**
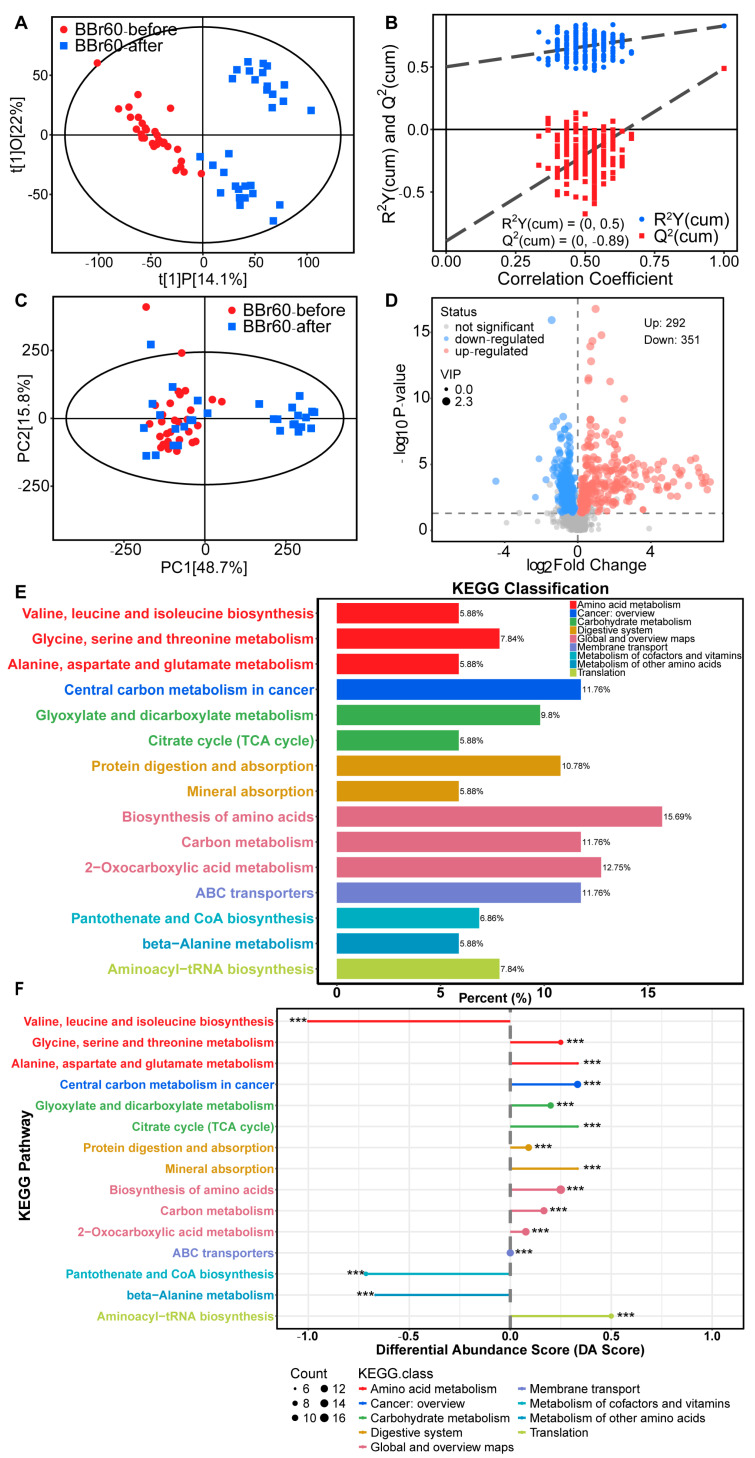
Serum metabolic profile between BBr60-after and BBr60-before groups. (**A**) The OPLS-DA scores of BBr60 vs. placebo group. (**B**) The OPLS-DA permutation test in BBr60 vs. placebo group. (**C**) The PCA scores of BBr60 vs. placebo group. (**D**) The volcano plot of placebo vs. BBr60 group. Significantly upregulated metabolites are represented by red points, and significantly downregulated metabolites and nonsignificant different ones are represented by blue or gray points, respectively. (**E**) KEGG classification plot for BBr60 vs. placebo group. (**F**) Differential abundance score plot for BBr60 vs. placebo group, *** *p* < 0.001.

**Figure 3 ijms-25-10871-f003:**
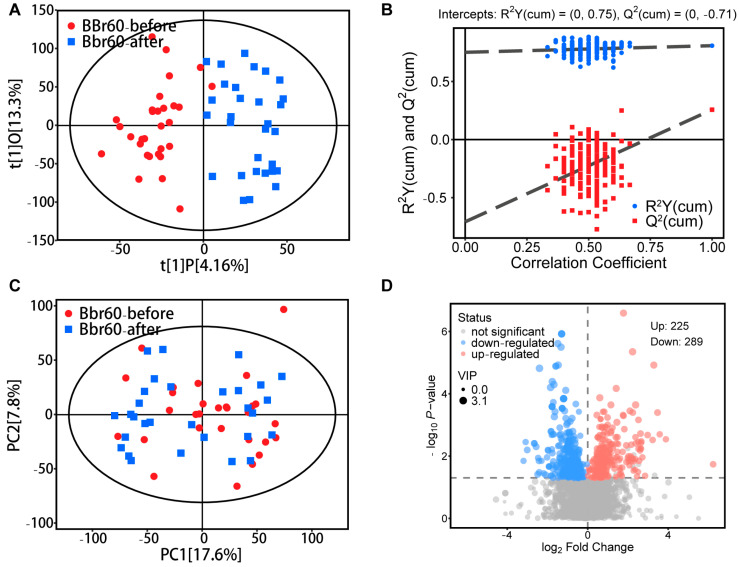
Fecal metabolic profile between BBr60-after and BBr60-before groups. (**A**) The OPLS-DA scores of BBr60 vs. placebo group. (**B**) The OPLS-DA permutation test in BBr60 vs. placebo group. (**C**) The PCA scores of BBr60 vs. placebo group. (**D**) The volcano plot of placebo vs. BBr60 group. Significantly upregulated metabolites are represented by red points, and significantly downregulated metabolites and nonsignificant different ones are represented by blue or gray points, respectively.

**Figure 4 ijms-25-10871-f004:**
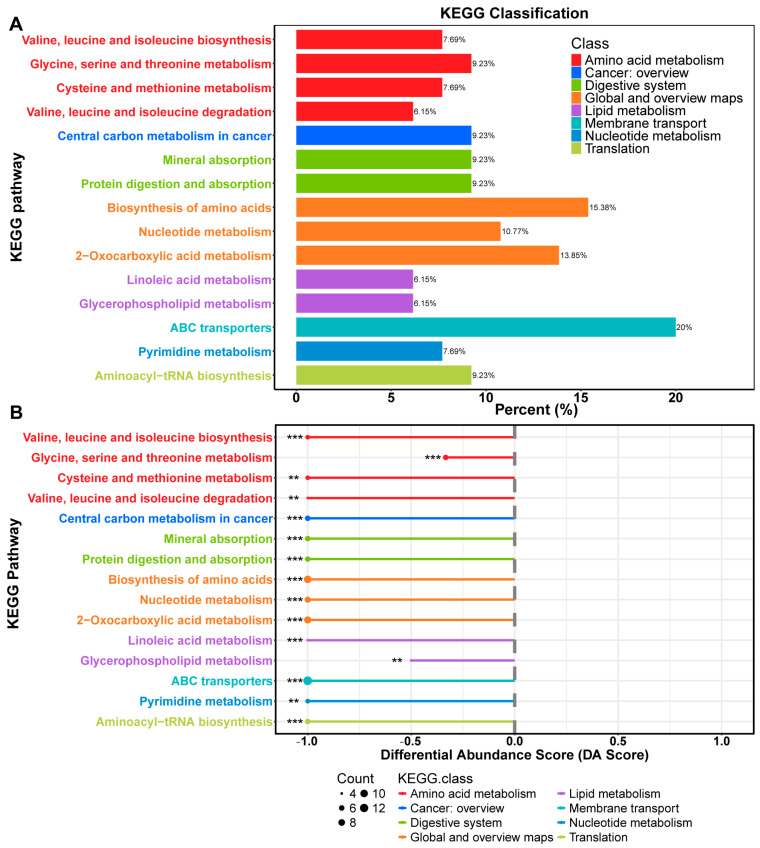
Fecal metabolic pathways between BBr60-after and BBr60-before groups. (**A**) KEGG classification plot for BBr60-after vs. BBr60-before groups. (**B**) Differential abundance score plot for BBr60-after vs. BBr60-before groups, ** *p* < 0.01, *** *p* < 0.001.

**Figure 5 ijms-25-10871-f005:**
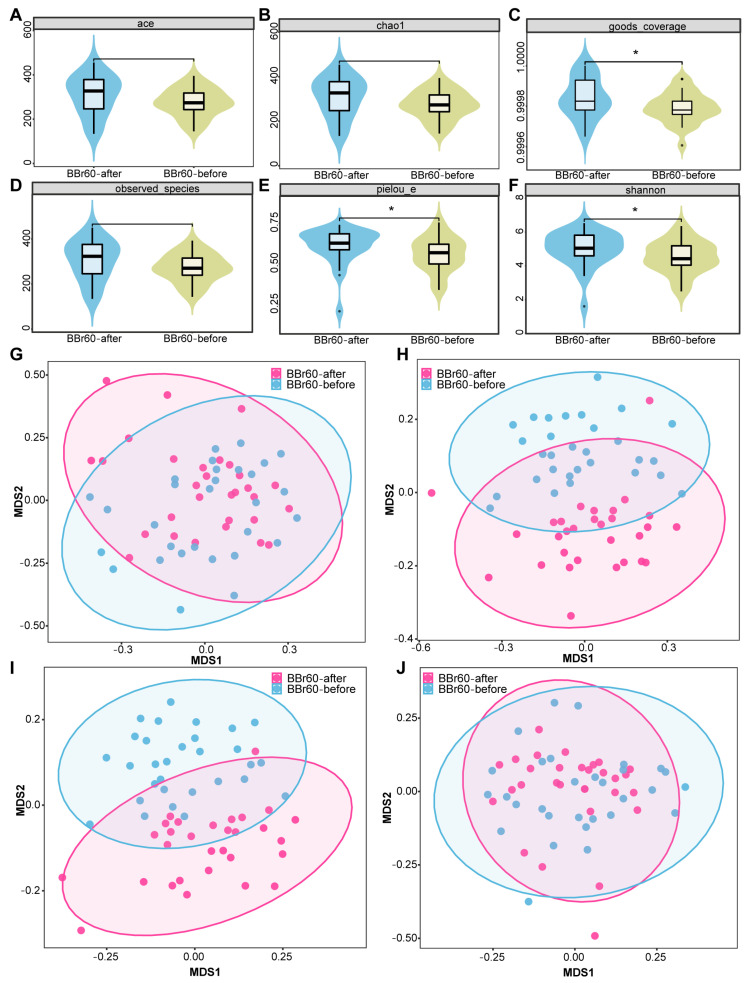
α and β diversity analysis of BBr60-after and BBr60-before group based on ace (**A**), chao1 (**B**), goods_coverage (**C**), observed_otus (**D**), pielou-e (**E**), Shannon (**F**), bray_curtis_distance (**G**), jaccard_distance (**H**), unweighted_unifrac_distance (**I**) weighted_unifrac (**J**) between BBr60, and placebo groups in the 12th week, * *p* < 0.05.

**Figure 6 ijms-25-10871-f006:**
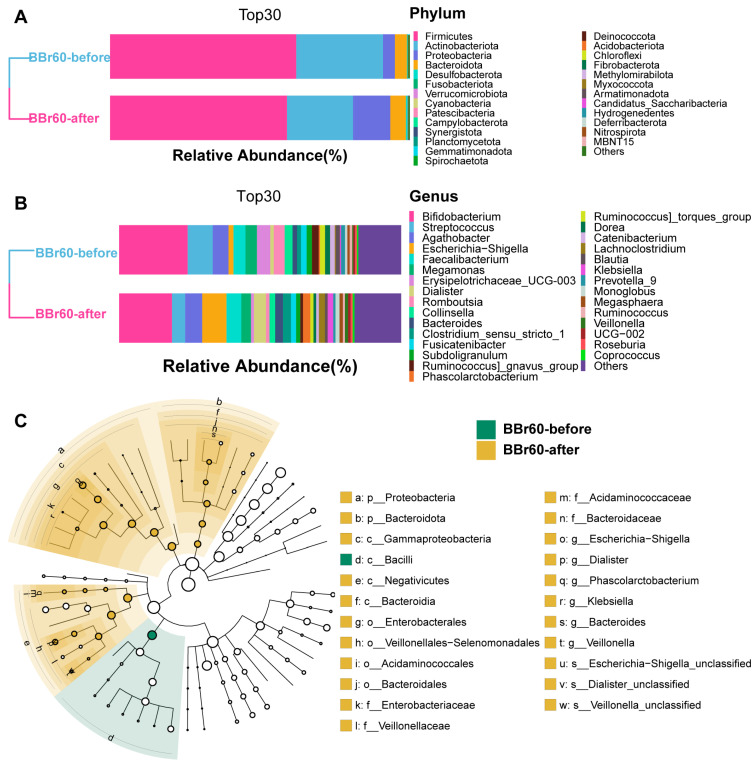
Bacterial compositions of BBr60-after and BBr60-before groups. (**A**) Composition of intestinal microbiota in the three groups at phylum level. (**B**) Composition of intestinal microbiota in the three groups at genus level. (**C**) Histogram of linear discriminant analysis (LDA) value distribution of intestinal microflora.

**Figure 7 ijms-25-10871-f007:**
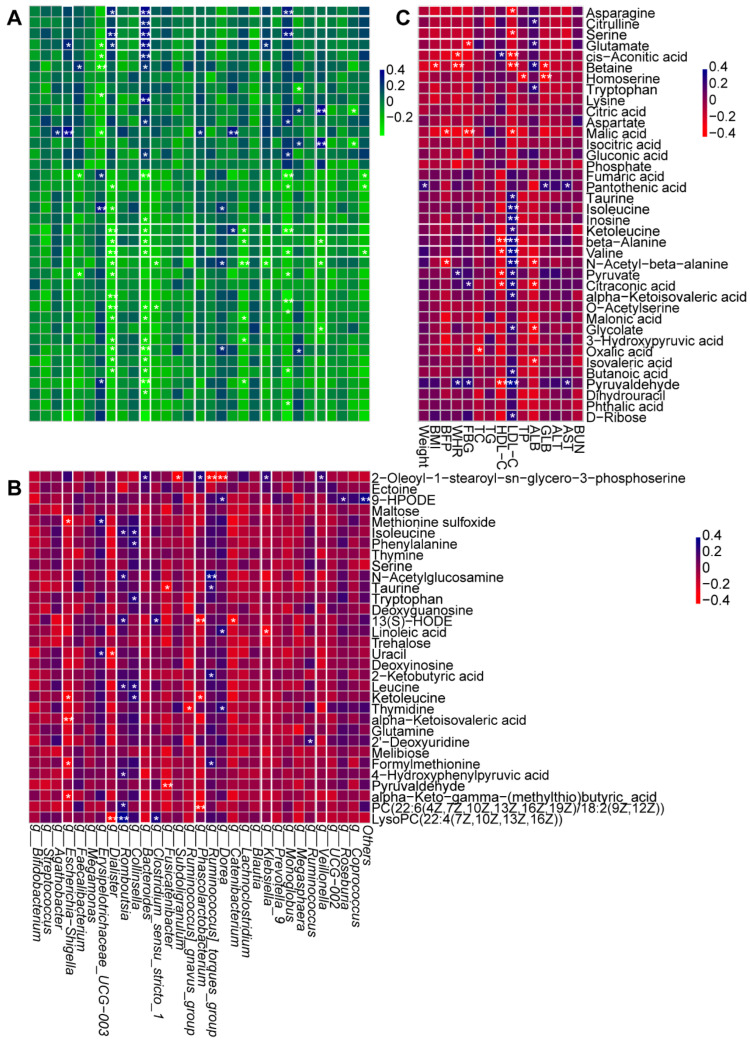
Correlation analysis of vital metabolites, intestinal bacteria, and clinic indexes before and after BBr60 intervention in overweight or obesity. (**A**) correlation analysis between serum metabolites associated with the top 15 changed metabolic pathways and with the top 30 intestinal bacteria before and after BBr60 intervention in overweight or obesity. The R values are represented by gradient colors, where blue and green cells indicate positive and negative correlations, respectively; * *p* < 0.05 and ** *p* < 0.01. (**B**) Correlation analysis between serum metabolites associated with the top 15 changed metabolic pathways and with the top 30 intestinal bacteria before and after BBr60 intervention in overweight or obesity. The R values are represented by gradient colors, where blue and red cells indicate positive and negative correlations, respectively; * *p* < 0.05 and ** *p* < 0.01. (**C**) Correlation analysis between serum metabolites associated with the top 15 changed metabolic pathways and clinic indexes in overweight or obesity. The R values are represented by gradient colors, where blue and red cells indicate positive and negative correlations, respectively; * *p* < 0.05 and ** *p* < 0.01. Abbreviations: BMI, body mass index; BFP, body fat percentage; WHR, waist-to-hip ratio; FBG, fasting blood glucose; TC, total cholesterol; TG, triglyceride; HDL-C, high-density lipoprotein cholesterol; LDL-C, low-density lipoprotein cholesterol; TP, total protein; ALB, albumin; GLB, globular proteins; ALT, alanine aminotransferase; AST, aspartate aminotransferase; BUN, blood urea nitrogen.

**Figure 8 ijms-25-10871-f008:**
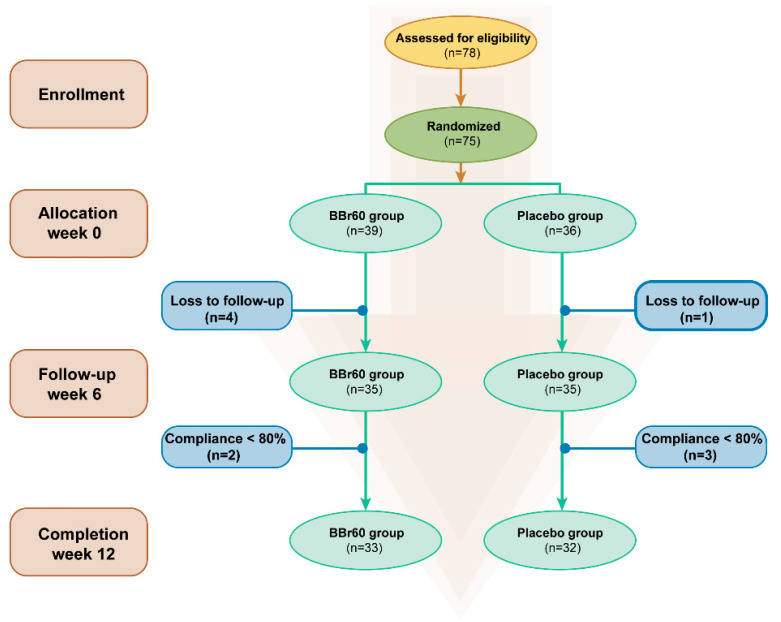
Flowchart of the study selection.

**Table 1 ijms-25-10871-t001:** Baseline characteristics in the probiotics and placebo groups.

Project	Unit	BBr60 (33)	Placebo (32)	*p*-Value
**Woman**	-	24 (72.7%)	19 (59.4%)	0.255
**Man**	-	9 (27.3%)	13 (40.6%)	
**Age**	year	27.88 ± 8.65	30.38 ± 8.45	0.150
**Weight**	kg	90.86 ± 10.45	93.56 ± 12.04	0.251
**BMI**	kg/m^2^	30.80 ± 3.21	31.96 ± 2.95	0.068
**BFP**	%	36.57 ± 6.77	38.91 ± 5.69	0.203
**WHR**	%	0.99 ± 0.05	1.01 ± 0.05	0.500
**FBG**	mg/dL	6.27 ± 0.87	5.87 ± 0.45	0.120
**TC**	mg/dL	4.31 ± 0.90	4.82 ± 0.94	0.058
**TG**	mg/dL	2.13 ± 1.78	1.62 ± 0.71	0.345
**HDL-C**	mg/dL	1.15 ± 0.27	1.29 ± 0.31	0.484
**LDL-C**	mg/dL	2.26 ± 0.63	2.62 ± 0.62	0.970
**ALT**	IU/L	39.27 ± 26.55	41.53 ± 21.84	0.618
**AST**	IU/L	61.18 ± 58.70	49.56 ± 17.88	0.969
**TP**	g/L	74.45 ± 9.70	72.47 ± 3.46	0.787
**ALB**	g/L	47.76 ± 3.93	47.41 ± 3.07	0.905
**GLB**	g/L	26.70 ± 7.28	25.06 ± 3.05	0.697
**A/G**	-	1.90 ± 0.41	1.92 ± 0.29	0.273
**TB**	mg/dL	13.58 ± 6.58	17.38 ± 14.83	0.453
**BUN**	mg/dL	4.48 ± 1.24	4.65 ± 1.26	0.846
**UA**	mg/dL	427.09 ± 76.69	433.59 ± 103.49	0.069
**CRE**	mg/dL	76.48 ± 14.80	71.16 ± 15.77	0.543

Abbreviations: BMI, body mass index; BFP, body fat percentage; WHR, waist-to-hip ratio; FBG, fasting blood glucose; TC, total cholesterol; TG, triglyceride; HDL-C, high-density lipoprotein cholesterol; LDL-C, low-density lipoprotein cholesterol; ALT, alanine aminotransferase; AST, aspartate aminotransferase; TP, total protein; ALB, albumin; GLB, globular proteins; A/G, albumin/globulin; TB, total bilirubin; BUN, blood urea nitrogen; UA, uric acid; CRE, creatinine.

**Table 2 ijms-25-10871-t002:** Changes in weight, BMI, body composition in the BBr60 and placebo groups after 12 weeks.

Variables	BBr60 (n = 33)			Placebo (n = 32)			*p*-Value
	Before (0 Week)	After (12 Week)	*p*-Value	Before (0 Week)	After (12 Week)	*p*-Value	
**Weight (kg)**	90.86 ± 10.45	86.19 ± 9.82	<0.0001	93.56 ± 12.04	90.74 ± 12.77	0.0006	0.114
**Weight (12–0 week)**	−4.67 ± 4.40			−2.82 ± 4.17			0.047
**BMI (kg/m^2^)**	30.80 ± 3.21	29.32 ± 3.63	<0.0001	31.96 ± 2.95	31.03 ± 3.49	0.0019	0.057
**BMI (12–0 week)**	−1.49 ± 1.37			−0.93 ± 1.55			0.135
**BFP (%)**	36.57 ± 6.77	34.54 ± 7.50	<0.0001	38.91 ± 5.69	37.11 ± 6.70	0.0003	0.150
**BFP (12–0 week)**	−2.03 ± 2.54			−1.80 ± 2.47			0.684
**WHR (%)**	0.99 ± 0.05	0.96 ± 0.04	<0.0001	1.01 ± 0.05	0.98 ± 0.05	<0.0001	0.149
**WHR (12-0 week)**	−0.036 ± 0.03			−0.038 ± 0.03			0.821

Abbreviations: BMI, body mass index; BFP, body fat percentage; WHR, where hearts rot.

**Table 3 ijms-25-10871-t003:** Changes in serum glucose and lipids in the BBr60 and placebo groups after 12 weeks.

Variables	BBr60 (n = 33)			Placebo (n = 32)			*p*-Value
	Before (0 Week)	After (12 Week)	*p*-Value	Before (0 Week)	After (12 Week)	*p*-Value	
**FBG, mg/dL**	5.87 ± 0.45	5.26 ± 0.57	<0.0001	6.27 ± 0.87	5.69 ± 0.86	<0.0001	0.0381
**TC, mg/dL**	4.31 ± 0.90	4.38 ± 0.75	0.618	4.82 ± 0.94	4.55 ± 0.90	0.0058	0.4181
**TG, mg/dL**	2.13 ± 1.78	1.98 ± 1.12	0.8566	1.62 ± 0.71	2.09 ± 1.34	0.0984	0.9870
**HDL-C, mg/dL**	1.15 ± 0.27	1.45 ± 0.28	<0.0001	1.29 ± 0.31	1.47 ± 0.26	0.0071	0.7589
**LDL-C, mg/dL**	2.26 ± 0.63	1.44 ± 0.52	<0.0001	2.62 ± 0.62	1.57 ± 0.53	<0.0001	0.3483

Abbreviations: FBG, fasting blood glucose; TC, total cholesterol; TG, triglyceride; HDL-C, high-density lipoprotein cholesterol; LDL-C, low-density lipoprotein cholesterol.

**Table 4 ijms-25-10871-t004:** Changes in liver and renal function in the BBr60 and placebo groups after 12 weeks.

Variables	BBr60 (n = 33)			Placebo (n = 32)			*p*-Value
	Before (0 Week)	After (12 Week)	*p*-Value	Before (0 Week)	After (12 Week)	*p*-Value	
**TP, g/L**	74.45 ± 9.70	72.85 ± 3.70	0.601	72.47 ± 3.46	73.19 ± 3.39	0.404	0.701
**ALB, g/L**	47.76 ± 3.93	50.09 ± 3.18	0.0003	47.41 ± 3.07	49.31 ± 2.01	0.0058	0.2037
**GLB, g/L**	26.70 ± 7.28	22.76 ± 3.85	0.0089	25.06 ± 3.05	23.94 ± 2.96	0.0499	0.170
**ALT, IU/L**	39.27 ± 26.55	26.03 ± 17.13	<0.0001	41.53 ± 21.84	27.22 ± 15.69	<0.0001	0.631
**AST, IU/L**	61.18 ± 58.70	38.27 ± 17.75	0.0002	49.56 ± 17.88	39.59 ± 17.49	0.0089	0.636
**BUN, mg/dL**	4.48 ± 1.24	4.19 ± 0.83	0.219	4.65 ± 1.26	4.73 ± 1.24	0.418	0.093

Abbreviations: ALT, alanine aminotransferase; AST, aspartate aminotransferase; TP, total protein; ALB, albumin; GLB, globular proteins; BUN, blood urea nitrogen.

## Data Availability

Data described in the manuscript, code book, and analytic code will be made available upon request pending application and approval.

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
