# Peer review of "Gut Microbiome and Metabolome Alterations in Overweight or Obese Adult Population after Weight-Loss Bifidobacterium breve BBr60 Intervention: A Randomized Controlled Trial"

_ijms, 2024, doi:10.3390/ijms252010871_

Round 1

Reviewer 1 Report

Comments and Suggestions for Authors

Dear Authors,

Congratulations for your research. It was an interesting read, however, it still requires several corrections and edits before publication can be considered. Please see my feedback below:

1. The following part does not belong to the introduction section. Please delete: "Seventy overweight/obese adult subjects were randomly assigned to receive either Bifidobacterium brevis BBr60 or a placebo for a duration of 12 weeks, along with brief dietary counseling emphasizing a total daily energy intake of 1800 kcal. Serum and fecal samples were collected to assess glucose and lipid levels, gut metabolomics, and microflora following treatment in order to evaluate the efficacy and mechanism of regulating metabolic disturbances in overweight/obese individuals. Ultimately, this study provides a scientific rationale for the dietary and clinical utilization of Bifidobacterium brevis BBr60 in the prevention and management of overweight/obesity.".

2. Rewrite the last paragraph of the introduction section where you should mention the study objectives and the study the hypothesis of the study.

3. The in-text citations are incorrectly used. Superscript is not necessary. Please remove it.

4. Include a PICO statement in the materials and methods in a short paragraph.

5. Include in the statistical analysis section how the normality of data was checked/tested.

6. Please format the tables according to journal's guidelines.

7. The following section can be removed and just mention this sentence somewhere else in the results: "3.5. Safety and tolerability - There was no serious adverse event in any of the groups during the study period, suggesting a favorable safety profile for BBr60.".

8. Include a short paragraph in the discussions section with study limitations. 

9. The last paragraph of the discussion section can be shortened or split.

10. Write a short paragraph about the clinical implications in medical practice of your findings.

Good luck!

Comments on the Quality of English Language

Minor corrections are necessary.

Author Response

Response to Reviewer 1 Comments

1. Summary

Thank you for your comments concerning our manuscript entitled “Gut microbiome and metabolome alterations in overweight or obese adult population after weight-loss Bifidobacterium brevis BBr60 intervention: a Randomized Controlled Trial” (Manuscript Number: ijms-3223309). Those comments are all valuable and very helpful for improving our paper. We have read comments carefully and made corrections. Revised portions were marked in red in the paper. The correction and response to the comments, which are also marked in red, are as follows:

2. Questions for General Evaluation

Reviewer’s Evaluation

Response and Revisions

Does the introduction provide sufficient background and include all relevant references?

Can be improved

Are all the cited references relevant to the research?

Can be improved

Is the research design appropriate?

Can be improved

Are the methods adequately described?

Can be improved

Are the results clearly presented?

Can be improved

Are the conclusions supported by the results?

Can be improved

3. Point-by-point response to Comments and Suggestions for Authors

Congratulations for your research. It was an interesting read, however, it still requires several corrections and edits before publication can be considered. Please see my feedback below:

Comments 1: The following part does not belong to the introduction section. Please delete: "Seventy overweight/obese adult subjects were randomly assigned to receive either Bifidobacterium brevis BBr60 or a placebo for a duration of 12 weeks, along with brief dietary counseling emphasizing a total daily energy intake of 1800 kcal. Serum and fecal samples were collected to assess glucose and lipid levels, gut metabolomics, and microflora following treatment in order to evaluate the efficacy and mechanism of regulating metabolic disturbances in overweight/obese individuals. Ultimately, this study provides a scientific rationale for the dietary and clinical utilization of Bifidobacterium brevis BBr60 in the prevention and management of overweight/obesity.".

Response 1: Thank you for your valuable suggestion, we have deleted this part, and adjusted the sentence to better fit the presentation in this section.

Comments 2: Rewrite the last paragraph of the introduction section where you should mention these study objectives and the study the hypothesis of the study.

Response 2: We sincerely thank you for careful reading, we have added the objectives and the hypothesis of the study in Line of 77-83.

Comments 3: The in-text citations are incorrectly used. Superscript is not necessary. Please remove it.

Response 3: We sincerely appreciate the valuable comment, we have adjusted the format of the references in the whole paper.

Comments 4: Include a PICO statement in the materials and methods in a short paragraph.

Response 4: We sincerely thank you for careful reading, “PICO statement”has been provided in Line of 91-104.

Comments 5: Include in the statistical analysis section how the normality of data was checked/tested.

Response 5: Thank you for your careful reading, the analytical method of normal distribution of the variables has been provided in Line of 210-211.

Comments 6: Please format the tables according to journal's guidelines.

Response 6: We sincerely appreciate the valuable comment, the formats of tables have been revised according to journal's guidelines. And the chart names are inserted in appropriate place in Line of 273 and 286.

Comments 7: The following section can be removed and just mention this sentence somewhere else in the results: "3.5. Safety and tolerability - There was no serious adverse event in any of the groups during the study period, suggesting a favorable safety profile for BBr60.".

Response 7: We sincerely appreciate the valuable comment, we have deleted this part and added this sentence in Line of 412-413.

Comments 8: Include a short paragraph in the discussions section with study limitations.

Response 8: Thank you for your suggestion, we have added this paragraph in Line of 573-580.

Comments 9: The last paragraph of the discussion section can be shortened or split.

Response 9: We sincerely appreciate the valuable comment. Considering that the regulatory mechanism analysis needs in-depth discussion, we have segmented the last paragraph without deleting it.

Comments 10: Write a short paragraph about the clinical implications in medical practice of your findings.

Response 10: Thank you for your suggestion, the research significance of the experimental results has been added in Line of 588-589.

Reviewer 2 Report

Comments and Suggestions for Authors

The manuscript explores the efficacy of Bifidobacterium brevis BBr60 administration in modulating metabolic profiles and gut microbiota in a group of 75 overweight/obese young persons. In my opinion, this is a well-conducted paper in a significant area of research regarding the relationship between obesity – metabolic dysfunction – and gut dysbiosis.

I have only a few suggestions for the authors:

- Obesity is one of the significant risk factors for diabetes or gut microbiota dysbiosis in type 2 diabetes. As overweight and obese patients were included in the study group, was the presence or absence of diabetes an exclusion criterion? Please clarify this aspect in the Study Design and Population (Section 2.2.)

- ’’All participants were instructed to reduce their daily energy intake by 1800 kcal and attended a nutrition information course covering topics such as the risks and causes of overweight and obesity, weight loss principles, dietary recommendations, and rest.’’ (lines 113-116). Please explain if this aspect (diet adaptation) has influenced the study results, such as blood glucose, total cholesterol, triglyceride or gut microbiota analysis.

- Kindly revise the main text. Sometimes, the letters have different sizes (please start with the manuscript title and the ‘’population’’  word).

- Kindly add a reference in line 73.

Author Response

For research article

Response to Reviewer 2 Comments

1. Summary

Thank you for your comments concerning our manuscript entitled “Gut microbiome and metabolome alterations in overweight or obese adult population after weight-loss Bifidobacterium brevis BBr60 intervention: a Randomized Controlled Trial” (Manuscript Number: ijms-3223309). Those comments are all valuable and very helpful for improving our paper. We have read comments carefully and made corrections. Revised portions were marked in red in the paper. The correction and response to the comments, which are also marked in red, are as follows:

2. Questions for General Evaluation

Reviewer’s Evaluation

Response and Revisions

Does the introduction provide sufficient background and include all relevant references?

Yes

Are all the cited references relevant to the research?

Yes

Is the research design appropriate?

Can be improved

Are the methods adequately described?

Can be improved

Are the results clearly presented?

Yes

Are the conclusions supported by the results?

Yes

3. Point-by-point response to Comments and Suggestions for Authors

The manuscript explores the efficacy of Bifidobacterium brevis BBr60 administration in modulating metabolic profiles and gut microbiota in a group of 75 overweight/obese young persons. In my opinion, this is a well-conducted paper in a significant area of research regarding the relationship between obesity – metabolic dysfunction – and gut dysbiosis.

I have only a few suggestions for the authors:

Comments 1: Obesity is one of the significant risk factors for diabetes or gut microbiota dysbiosis in type 2 diabetes. As overweight and obese patients were included in the study group, was the presence or absence of diabetes an exclusion criterion? Please clarify this aspect in the Study Design and Population (Section 2.2.)

Response 1: Thank you for your suggestion. None of the enrolled participants had diabetes, and corresponding remarks has been added in the Line of 113.

Comments 2: ’’All participants were instructed to reduce their daily energy intake by 1800 kcal and attended a nutrition information course covering topics such as the risks and causes of overweight and obesity, weight loss principles, dietary recommendations, and rest.’’ (lines 113-116). Please explain if this aspect (diet adaptation) has influenced the study results, such as blood glucose, total cholesterol, triglyceride or gut microbiota analysis.

Response 2: Thank you for your valuable comment. This is a vital part to explain the effect, the results of weight, BMI, BFP, WHR, FBG, blood lipid (TC, HDL-C, LDL-C), liver and renal function (ALB, ALT, AST) presented effective regulation (p < 0.05), but the effectiveness may be not absolutely attributed to dietary calorie restriction, and individual active physical exercise may also play an important role, but the specific exercise situation is not detailed statistics throughout the study. Therefore, further studies are necessary to investigate the effectiveness of BBr60 in mice, specifically isolating the variables of dietary calorie restriction and daily physical exercise.

Comments 3: Kindly revise the main text. Sometimes, the letters have different sizes (please start with the manuscript title and the ‘’population’’ word).

Response 3: Thank you for your comment. We have corrected the problems in Line of 4, 347, 396.

Comments 4: Kindly add a reference in line 73.

Response 4: We sincerely appreciate the valuable comment. The anti-inflammation and antioxidant properties of Bifidobacterium breve BBr60 have been found, but this work is published in the form of a Chinese patent, and the corresponding article is not published with English paper form. So, we didn't insert the corresponding reference in appropriated place.

This is the Chinese patent:

Bifidobacterium breve, which has the effects of repairing ultraviolet damage, alleviating inflammation and preventing skin photoaging, and the preparation method and application thereof, patent number: CN111235058A

Round 2

Reviewer 1 Report

Comments and Suggestions for Authors

The revised version satisfies all previous concerns. No additional suggestions.

Comments on the Quality of English Language

English quality is fine. Minor adjustments needed.